# Link Recommendation to Augment Influence Diffusion with Provable Guarantees

## ABSTRACT

Link recommendation systems in online social networks (OSNs), such as Facebook's "People You May Know", Twitter's "Who to Follow", and Instagram's "Suggested Accounts", facilitate the formation of new connections among users. This paper addresses the challenge of link recommendation for the purpose of social influence maximization. In particular, given a graph $G$ and the seed set $S$, our objective is to select $k$ edges that connect seed nodes and ordinary nodes to optimize the influence dissemination of the seed set. This problem, referred to as influence maximization with augmentation (IMA), has been proven to be NP-hard. In this paper, we propose an algorithm, namely AIS, consisting of an efficient estimator for augmented influence estimation and an accelerated sampling approach. AIS provides a $(1 - 1/e - \varepsilon)$-approximate solution with a high probability of $1 - \delta$, and runs in $O(k^2(m+n)\log(n/\delta)/\varepsilon^2 + k|E_C|)$ time assuming that the influence of any singleton node is smaller than that of the seed set. To the best of our knowledge, this is the first algorithm that can be implemented on large graphs containing millions of nodes while preserving strong theoretical guarantees. We conduct extensive experiments to demonstrate the effectiveness and efficiency of our proposed algorithm.

## CCS CONCEPTS

• **Mathematics of computing → Graph algorithms**; • **Information systems → Social networks**.

## KEYWORDS

Social Networks, Influence Maximization

**ACM Reference Format:**

Anonymous Author(s). 2023. Link Recommendation to Augment Influence Diffusion with Provable Guarantees. In *Proceedings of the ACM Web Conference 2024 (WWW '24)*. ACM, New York, NY, USA, 9 pages. https://doi.org/XXXXXXX.XXXXXXX

## 1 INTRODUCTION

Online social networks (OSNs) are becoming an increasingly powerful medium for disseminating useful content. Nowadays, individuals are seamlessly connected, forming intricate webs of relationships that facilitate the exchange of information, ideas, and opinions on an unprecedented scale. As a result, understanding and harnessing the

mechanisms behind influence spread have become crucial for various domains [13], ranging from political campaigns to marketing [10]. Viral marketing, in particular, has gained substantial traction in recent years as a cost-effective and efficient strategy to promote products, services, and ideas, which leverages the interconnectedness of social networks to facilitate one-to-one or many-to-many communication, enabling messages to spread rapidly and organically. By harnessing the power of social influence, viral marketing campaigns aim to create a cascade of user engagements through the word-of-mouth effect, leading to exponential growth in reach and impact.

An important scientific problem related to that is influence maximization (IM) [18], which aims to select a set of nodes in a social network as the sources of influence spread to maximize the expected number of influenced nodes. The seminal work of Kempe et al. [18] formulates this problem as a submodular optimization problem. Their results have sparked a whole line of research on the IM problem [2, 5–7, 11, 14, 15, 23, 31–33]. Such work normally assumes that the topology structure of the network does not change.

Another way to boost the influence spread in an OSN is by increasing the connectivity among users. Some OSN platforms like Twitter already use "people recommendations" to increase connectivity. However, most recommendation systems mainly focus on making relevant recommendations without an explicit effort towards augmenting information spread. For example, the "People You May Know" feature employs the Friend-of-Friend (FoF) algorithm [25], recommending the users that have the highest common friends with the target user receiving the recommendation. Other recommendation algorithms may recommend users whose profiles have substantial overlap with the receiver. However, simply recommending connections based on the number of mutual friends or similarity may not maximize the influence spread in a social network. The combination of link recommendation with information diffusion in OSNs opens up new opportunities for product marketing.

Recommending links allows us to tap into the network's inherent structure and dynamics, enabling us to identify meaningful connections that can facilitate influence diffusion specifically within the desired audience. Furthermore, combining the selection of influential users with link recommendations creates a synergistic effect. The influential users can initiate the spread of influence, while the recommended links enhance and amplify the diffusion process of the target group. In addition, selecting seed users usually bring high costs in both expense and time. Thus, adding new connections into the network is a more economic approach to amplify the influence spread than selecting new seed users.

In contrast to the previous works of influence maximization where the network topology remains unchanged, we are interested in recommending links that can augment the social influence of a target group of users. More specifically, we aim to suggest a fixed

number of new connections to a subset of users to maximize the expected influence spread of them. This problem is known as influence maximization with augmentation (IMA), which is first proposed by D'Angelo et al. [9]. They show that this problem is NP-hard and the objective function is submodular under the Independent Cascade (IC) model. As a solution, they utilize a greedy approach with $(1 - 1/e - \varepsilon)$-approximations [27], which unfortunately suffers from a serious scalability issue.

Specifically, an important element of the greedy algorithm is to compute the influence spread for the seed set, which is #P-hard [5]. D'Angelo et al. [9] propose to estimate the influence spread via Monte-Carlo simulations, where the resultant algorithm is called MC-Greedy. To obtain accurate estimation of the influence spread, a large number of Monte-Carlo simulations are needed, incurring significant computational overheads. Inspired by the groundbreaking reverse influence sampling (RIS) method [2], we propose a RIS-based algorithm with two newly developed innovative techniques, including an efficient estimator and an accelerated sampling approach tailored for the IMA problem. To the best of our knowledge, this is the first algorithm that can be applied to large-scale networks with theoretical guarantees.

**Contributions.** In this paper, we study the IMA problem and present AIS that overcomes the deficiencies of MC-Greedy. We illustrate the connection between the influence estimation problem and the IMA problem, which helps us to compute the marginal gain of a given candidate edge more efficiently by RIS. With this important observation, we propose an approximation algorithm for the IMA problem with two acceleration techniques based on the nature of the IMA problem. Theoretically, our algorithm achieves an approximation ratio of $(1 - 1/e - \varepsilon)$ with a high probability of $1 - \delta$ and runs in $O(\frac{k^2(m+n)\log(kn/\delta)}{\varepsilon^2} + k|E_C|)$ time when every singleton node's influence is smaller than the seed set. Practically, with extensive experiments on various datasets, our algorithm outperforms the baselines, and is the first method that can be applied to large datasets without any compromise of theoretical assurance. In summary, our contributions are as follows:

(1) We develop an efficient estimator for augmented influence estimation by building an connection between the IMA problem and the influence estimation problem, and propose novel techniques for accelerating sampling process.

(2) We propose AIS based on our sampling approach that returns a $(1 - 1/e - \varepsilon)$-approximate solution with probability $1 - \delta$ for the IMA problem, and runs in $O(k^2(m + n)\log(kn/\delta)/\varepsilon^2 + k|E_C|)$ time if any singleton node's influence is less than that of the seed set.

(3) We perform extensive experiments on various real-world datasets with up to millions of nodes and billions of edges and demonstrate the effectiveness and efficiency of our algorithm.

**Organization.** The rest of the paper is organized as follows. Section 2 reviews the related work. Section 3 gives the problem definition and introduces some necessary preliminaries. Section 4 devises the design of the AIS algorithm. Section 5 provides theoretical analysis. Section 6 evaluates the effectiveness and efficiency of our method. Finally, Section 7 concludes this work.

**Table 1: Frequently used notations**

| Notation | Description |
|---|---|
| $G$ | A social network |
| $V, E$ | The set of nodes and edges, respectively |
| $n, m$ | The number of nodes and edges in G |
| $E_C$ | Candidate edges set |
| $k$ | The number of edges to be selected |
| $S, A$ | The seed set, and the edge set |
| $\sigma(A, S)$ | The augmented influence spread of $S$ after adding the edge set $A$ |
| $R, \mathcal{R}$ | A random RR set, a collection of RR sets |
| $\Lambda_{\mathcal{R}}(S)$ | The number of RR sets in $\mathcal{R}$ that intersects with $S$ |

## 2 RELATED WORK

### 2.1 Influence Maximization

In 2003, Kempe et al. [18] publish the first algorithmic study on the influence maximization problem. They show that this problem under the independent cascade (IC) model is NP-hard, and propose a greedy algorithm to approximate the solution with a factor of $(1 - 1/e - \varepsilon)$. The key idea is to selects the node that gives the most significant increment of expected influence spread estimated via Monte-Carlo simulations [26]. Later, there have been several attempts [1, 5–7, 11, 12, 17, 23, 28] to improve the efficiency.

In 2014, Borgs et al. [2] make a significant breakthrough by presenting a near-linear time algorithm under the IC model. They introduce the concept of reverse influence sampling (RIS), which transforms the IM problem into a maximum coverage problem and provides a $(1 - 1/e - \varepsilon)$-approximate solution with high probability. Subsequently, Tang et al. [33] point out the shortcomings of Borgs's algorithm and then propose TIM to enhance the efficiency of RIS in practice. Tang et al. [32] propose IMM by adopting a martingale approach, which reduces the number of RR sets to deliver a guaranteed result. Later, [31] develop an RIS-based online processing algorithm and a novel approach to compute empirical guarantees, enabling early stopping of the RIS-based algorithms. In addition to the number of RR sets, researchers have also endeavored to design more sophisticated sampling techniques. Guo et al. [14] employ subset sampling to reduce the time complexity, referred to as SUBSIM. Zhu et al. [36] propose 2-hop+ sampling to boost estimation.

### 2.2 Diffusion-Aware Link Manipulation

There has been a plethora of research on diffusion-aware link manipulation in social networks [3, 8, 9, 16, 20, 21, 30]. This category of work aims to optimize information diffusion-related functions by adding or removing a limited number of edges in the social network. For instance, Khalil et al. [20] consider two optimization problems, i.e., adding edges to maximize influence spread and deleting edges to minimize influence spread. Yang et al. [34, 35] investigate how to add a limited number of edges from a candidate set to maximize the seed's influence in a directed acyclic graph under the IC model. Huang et al. [16] examine the problem of selecting edges from the original network to maximize the influence of specific seeds via these edges. Sun et al. [30] propose algorithms to identify fragile nodes and edges to attack to reduce the influence spread.

Different from these problems, we are interested in the IMA problem under the IC model proposed by D'Angelo et al. [9], which aims to select $k$ edges incident to the seed set to maximize the influence of the seed set $S$. The objective function is a monotone and submodular [9], and a Monte-Carlo-simulation-based greedy approach can achieve an approximation ratio of $(1 - 1/e - \varepsilon)$ [27]. Coró et al. [8] study the IMA problem under the LT model and show that the objective function is modular, indicating that a Monte-Carlo-simulation-based greedy approach provides $(1 - \varepsilon)$-approximate solution. However, such Monte-Carlo-simulation-based methods [8, 9] suffer from prohibitive computation overheads to provide theoretical guarantees. We tackle this issue via a non-trivial adoption of RIS.

## 3 PRELIMINARIES

In this section, we give a formal definition of the influence maximization with augmentation (IMA) problem, and introduce the greedy framework as well as the technique of reverse influence sampling (RIS). Notations that are frequently used in this paper are given in Table 1 for ease of reference.

### 3.1 Problem Definition

Let $G$ be a social network with a node set $V$ (representing users) and a directed edge set $E$ (representing connections among users), with $|V| = n$ and $|E| = m$. Each directed edge $\langle u, v \rangle \in E$ of $G$ is associated with a propagation probability $p_{u,v} \in [0, 1]$, representing the probability that $u$ can influence $v$.

In this paper, we study the basic and widely adopted independent cascade (IC) model. Initially, at timestamp 0, the selected seed nodes are activated, while all others are inactive. When a node first becomes activated at timestamp $i$, it has one single chance to activate its inactive neighbors with probability $p_{u,v}$ at timestamp $i+1$, and this node will remain active till the end of the propagation process. The diffusion process terminates when no more nodes in the graph can be activated.

Let $\sigma(S)$ denote the expected number of nodes activated by a seed set $S$ in graph $G$, which is also called the *expected spread* of $S$. The traditional IM problem asks for a set $S$ of seed nodes with the largest expected spread $\sigma(S)$. In this paper, we study the influence maximization with augmentation (IMA) problem [9] that aims to add $k$ edges from a candidate edge set $E_C$ to the original edge set $E$ to augment the expected spread of a given seed set $S$ as much as possible. Denote by $G(A)$ the augmented graph with a set $A$ of edges added to the base graph $G$, and by $\sigma(A, S)$ the augmented influence spread of $S$ on $G(A)$. For notational simplicity, let $\sigma(S) = \sigma(\emptyset, S)$. The IMA problem is formally defined as follows.

DEFINITION 3.1 (IMA [9]). *Given a graph $G = (V, E)$, a seed set $S$, a candidate edge set $E_C \subseteq (S \times V) \setminus E$, and a budget $k$, the influence maximization with augmentation (IMA) problem asks for $k$ edges in $E_C$ that maximizes the augmented influence spread of $S$. That is,*

$$A^* = \underset{A \subseteq E_C, |A| \leq k}{\arg\max} \sigma(A, S). \tag{1}$$

In what follows, we show that the IMA problem under the IC model is NP-hard to approximate within a factor greater than $1 - 1/e$. For a graph $G$, we consider an isolated node $u$ as the seed set and $p_{u,v} = 1$ for each $v \in V \setminus \{u\}$. Then, selecting $k$ edges to maximize

---

**Algorithm 1** Greedy

**Input:** $G, S, E_C, k$;
**Output:** An size-$k$ edge set $A$;
1: $A \leftarrow \emptyset$;
2: **for** $i \leftarrow 1$ to $k$ **do**
3:     $e^* \leftarrow \arg\max_{e \in E_C \setminus A} \widehat{\sigma}(A \cup \{e\}, S)$;
4:     $A \leftarrow A \cup \{e^*\}$;
5: **return** $A$;

---

$\sigma(A, \{u\})$ is equivalent to selecting a set $T$ of $k$ nodes in $V \setminus \{u\}$ to maximize $\sigma(T)$, i.e., the IM problem. It is known that the IM problem is NP-hard to approximate within a factor of $(1 - 1/e + \varepsilon)$ for any $\varepsilon$ under the IC model [18, 19, 29]. As a consequence, such a hardness result also applies to the IMA problem.

### 3.2 Greedy Framework

D'Angelo et al. [9] show that the objective function $\sigma(A, S)$ is submodular with respect to the edge set $A$. Therefore, the greedy algorithm can achieve an approximation ratio of $(1 - 1/e)$ [27]. Moreover, the computation of influence spread is #P-hard [5]. Chen et al. [4] show that the greedy algorithm on an accurate estimate of a monotone submodular function achieves an approximation ratio of $(1 - 1/e - \varepsilon)$.

LEMMA 3.2 ([4]). *Let $A^* = \arg\max_{|A| \leq k} f(A)$ be the set maximizing $f(A)$ among all sets with size at most $k$, where $f$ is monotone and submodular, and $f(\emptyset) = 0$. For any $\varepsilon > 0$ and any $0 < \lambda \leq \frac{\varepsilon/k}{2 + \varepsilon/k}$, if a set function $\hat{f}$ is a multiplicative $\lambda$-error estimate of set function $f$, the output $A^g$ of the greedy algorithm on $\hat{f}$ guarantees*

$$f(A^g) \geq (1 - 1/e - \varepsilon) f(A^*). \tag{2}$$

By Lemma 3.2, if $\widehat{\sigma}(A, S)$ is a multiplicative $\lambda$-error estimate of $\sigma(A, S)$, Greedy (i.e., Algorithm 1) returns a $(1 - 1/e - \varepsilon)$-approximate solution for the IMA problem. A naive way is to instantiate Greedy using Monte-Carlo simulations for estimating $\sigma(A, S)$, referred to as MC-Greedy. However, as analyzed in Section 5, MC-Greedy has a high time complexity of $O(\frac{k^3 mn|E_C| \log(|E_C|/\delta)}{\varepsilon^2})$, where $\delta \in (0, 1)$ is a predefined threshold on failure probability.

### 3.3 Reverse Influence Sampling

Borgs et al. [2] propose a novel idea of reverse sampling for the IM problem, referred to as reverse influence sampling (RIS). It uses sketch samples called *random reverse reachable (RR) sets* to estimate the expected spread of a seed set. A random RR set can be constructed in two steps.

(1) Select a node $v$ from $V$ uniformly at random.
(2) Collect a sample set $R$ of the nodes in $V$, such that for any $u \in V$, the probability that it appears in $R$ equals the probability that $u$ can activate $v$ in an influence propagation process.

A key observation is established as follows.

LEMMA 3.3 ([2]). *For any seed set $S \in V$ and a random RR set $R$,*

$$\sigma(S) = n \cdot \Pr[S \cap R \neq \emptyset].$$

According to Lemma 3.3, the expected spread of any seed set $S$ can be estimated by random RR sets. Specifically, given a set $\mathcal{R}$ of random RR sets, we say that a RR set $R \in \mathcal{R}$ is *covered* by a node set $S$ if $R \cap S \neq \emptyset$. Denote by $\Lambda_{\mathcal{R}}(S)$ the number of RR sets that are covered by $S$, referred to as the coverage of $S$ in $\mathcal{R}$. Then, $\frac{n}{|\mathcal{R}|} \Lambda_{\mathcal{R}}(S)$ is an unbiased estimate of $\sigma(S)$. On the basis of RIS, Borgs et al. [2] propose a general framework for IM consisting of two steps. That is, we first (i) generate a set $\mathcal{R}$ of random RR sets, and then (ii) identify a node set $S^*$ with the maximum coverage in $\mathcal{R}$ (via a standard greedy method). Utilizing the RIS technique, the state-of-the-art IM algorithms [14, 15, 31, 32] with reduced computation overheads have been proposed that ensure $(1 - 1/e - \varepsilon)$-approximations.

Intuitively, we may leverage the power of RIS for IMA. In particular, addressing IMA requires an effective and efficient estimation of $\sigma(A, S)$ for various edge sets $A$'s under a given node set $S$. Note that for IM, a random RR set is constructed independently from $S$ for estimating the expected spread $\sigma(S)$, since graph $G$ is static. In contrast, for IMA, generating a random RR set for estimating the augmented influence spread $\sigma(A, S)$ relies on the newly added edge set $A$, as the augmented graph $G(A)$ changes. Therefore, the key challenge for IMA via RIS lies in an efficient way to generate random RR sets that can estimate $\sigma(A, S)$ accurately with respect to different $A$.

# 4 AN EFFICIENT APPROXIMATION ALGORITHM FOR IMA

We leverage the greedy framework, i.e., Algorithm 1, that achieves the best approximation ratio for the IMA problem. The core procedure of Algorithm 1 is to select the edge $e$ with the largest estimated $\widehat{\sigma}(A \cup \{e\}, S)$ of the augmented influence spread $\sigma(A \cup \{e\}, S)$ (Line 3). In this section, we first propose an efficient estimator built upon RR sets that can estimate $\widehat{\sigma}(A \cup \{e\}, S)$ with sufficient accuracy considering all $e$'s simultaneously. Based on such an estimator, we devise a scalable algorithm, namely AIS, that provides an approximation of $(1 - 1/e - \varepsilon)$ for IMA.

## 4.1 An Efficient Estimator via RR Sets

When examining each edge $e$ in the greedy selection procedure, since the augmented graph $G(A \cup \{e\})$ varies, we should generate distinct RR sets for estimating $\sigma(A \cup \{e\}, S)$ as mentioned in Section 3.3, incurring prohibitive overheads. To tackle this issue, we propose a novel unbiased estimate of $\sigma(A \cup \{e\}, S)$ that can break the dependency between RR sets generation and the candidate edge $e$ to be examined. Specifically, we express $\sigma(A \cup \{e\}, S)$ as a combination of $\sigma(A, S \cup \{v\})$ and $\sigma(A, S)$. Since the augmented graph $G(A)$ is independent of $e$, the RR sets for estimating $\sigma(A, S \cup \{v\})$ and $\sigma(A, S)$ can be generated independently from $e$.

**LEMMA 4.1.** *Under the IC model, for any edge $e = \langle u, v \rangle \in E_C \setminus A$ associated with a propagation probability $p_{u,v}$, we have*

$$\sigma(A \cup \{e\}, S) = p_{u,v} \cdot \sigma(A, S \cup \{v\}) + (1 - p_{u,v}) \cdot \sigma(A, S).$$

**PROOF.** Consider the live-edge graph expression $g$ of $G$ such that $g$ is generated by independently flipping a coin of bias $p_{u,v}$ for each edge $\langle u, v \rangle \in E$ to decide whether the edge is live or blocked, referred to as $g \sim G$. Let $I_g(S)$ be the influence spread of $S$ on $g$,

i.e., the number of nodes that are reachable from $S$ on the sample outcome $g$. Then, the expected spread $\sigma(A, S)$ can be written as

$$\sigma(A, S) = \sum_{g \sim G(A)} \left( \Pr[g] \cdot I_g(S) \right).$$

Similarly,

$$\sigma(A \cup \{e\}, S) = \sum_{g' \sim G(A \cup \{e\})} \left( \Pr[g'] \cdot I_{g'}(S) \right).$$

According to the distribution of $g$ and $g'$, let

$$g' = \begin{cases} g \cup \{e\}, & \text{if } e \text{ is live with a probability of } p_{u,v}, \\ g, & \text{if } e \text{ is blocked with a probability of } 1 - p_{u,v}. \end{cases}$$

Then, we can rewrite $\sigma(A \cup \{e\}, S)$ as

$$\sigma(A \cup \{e\}, S) = \sum_{g \sim G(A)} \Pr[g] \cdot \left( p_{u,v} \cdot I_{g \cup \{e\}}(S) + (1 - p_{u,v}) \cdot I_g(S) \right).$$

When edge $e$ is live in $g'$, $v$ is activated by $S$, which can be viewed as a seed node, indicating that $I_{g \cup \{e\}}(S) = I_g(S \cup \{v\})$. Therefore,

$$\sigma(A \cup \{e\}, S) = \sum_{g \sim G(A)} \Pr[g] \cdot \left( p_{u,v} \cdot I_g(S \cup \{v\}) + (1 - p_{u,v}) \cdot I_g(S) \right)$$

$$= p_{u,v} \cdot \sigma(A, S \cup \{v\}) + (1 - p_{u,v}) \cdot \sigma(A, S).$$

This completes the proof. □

Based on Lemma 4.1, to perform greedy selection, we can estimate $\sigma(A \cup \{e\}, S)$ via a combination of $\widehat{\sigma}(A, S \cup \{v\})$ and $\widehat{\sigma}(A, S)$. Note that under given $A$ and $S$, an edge $e$ maximizing $p_{u,v} \cdot \left( \widehat{\sigma}(A, S \cup \{v\}) - \widehat{\sigma}(A, S) \right)$ also maximizes $p_{u,v} \cdot \widehat{\sigma}(A, S \cup \{v\}) + (1 - p_{u,v}) \cdot \widehat{\sigma}(A, S)$, since the additive term $\widehat{\sigma}(A, S)$ in the latter is independent of $e$. For convenience, in each iteration, we select the edge with the largest value of $p_{u,v} \cdot \left( \widehat{\sigma}(A, S \cup \{v\}) - \widehat{\sigma}(A, S) \right)$. Given a fixed node set $S$, for any node $v$ and any RR set $R$, let $\mathbb{I}_R(v)$ be an indicator function such that

$$\mathbb{I}_R(v) = \begin{cases} 1, & R \text{ is covered by } v \text{ but not by } S, \\ 0, & \text{otherwise.} \end{cases} \tag{3}$$

In addition, given a set $\mathcal{R}$ of random RR sets, denote by $\Delta_{\mathcal{R}}(v)$ the number of RR sets in $\mathcal{R}$ covered by $v$ but not covered by $S$, i.e., $\Delta_{\mathcal{R}}(v) = \sum_{R \in \mathcal{R}} \mathbb{I}_R(v) = \Lambda_{\mathcal{R}}(S \cup \{v\}) - \Lambda_{\mathcal{R}}(S)$. Then, for a set $\mathcal{R}$ of random RR sets generated on the augmented graph $G(A)$, $\frac{n}{|\mathcal{R}|} \Delta_{\mathcal{R}}(v)$ is an unbiased estimate of $\sigma(A, S \cup \{v\}) - \sigma(A, S)$. Putting it together yields that $\frac{n p_{u,v}}{|\mathcal{R}|} \Delta_{\mathcal{R}}(v)$ is an unbiased estimate of $\sigma(A \cup \{e\}, S) - \sigma(A, S)$.

## 4.2 Accelerating RR Sets Generation

For each RR set $R$, by definition, $\mathbb{I}_R(\cdot) = 0$ if $R$ is covered by $S$. This implies that during the generation process of $R$, if any node in $S$ is added to $R$, we can stop the process immediately to produce a truncated RR set $R'$. Apparently, $R'$ is also covered by $S$. As a result, we still have $\mathbb{I}_{R'}(\cdot) = \mathbb{I}_R(\cdot) = 0$ while generating $R'$ accelerates the sampling process.

Moreover, in each iteration, RR sets are generated based on current augmented graph $G(A)$. After selecting a new edge $e$, the graph is further augmented from $G(A)$ to $G(A \cup \{e\})$. A naive way is to generate new RR sets from scratch for each iteration, which is time consuming. Instead, we propose to update the RR sets

---

**Algorithm 2** AIS

---

**Input:** $G, S, E_C, k, \varepsilon, \delta$;
**Output:** An size-$k$ edge set $A$;

1: $\delta \leftarrow \frac{\delta}{k|E_C|}, \lambda \leftarrow \frac{\varepsilon/k}{2 + \varepsilon/k}$;

2: Generate random RR sets until $\Lambda_{\mathcal{R}}(S) \geq \frac{2(1+\lambda)(1+\frac{\lambda}{3})\log(2/\delta)}{\lambda^2}$;

3: Initialize $\Delta_{\mathcal{R}}(v) \leftarrow 0$ for each $v \in V$;

4: **for** each $R \in \mathcal{R}$ **do**

5:     $\Pi(R) \leftarrow 1$;    ▷ *Indicate whether $R$ is covered by $S$*

6:     **if** $R$ is not covered by $S$ **then**

7:         $\Pi(R) \leftarrow 0$;

8:         **for** each $v \in R$ **do**

9:             $\Delta_{\mathcal{R}}(v) \leftarrow \Delta_{\mathcal{R}}(v) + 1$;

10: $A \leftarrow \emptyset$;

11: **for** $i \leftarrow 1$ to $k$ **do**

12:     $e^* \leftarrow \arg\max_{e \in E_C \setminus A} p_{u,v} \cdot \Delta_{\mathcal{R}}(v)$;    ▷ *Denote $e^* = \langle u^*, v^* \rangle$*

13:     $A \leftarrow A \cup \{e^*\}$;

14:     **for** each $R \in \mathcal{R}$ **do**    ▷ *Update RR sets in a soft way*

15:         **if** $\Pi(R) = 0$ & $v^* \in R$ & $U(0, 1) \leq p_{u^*, v^*}$ **then**

16:             $\Pi(R) \leftarrow 1$;

17:             **for** each $v \in R$ **do**

18:                 $\Delta_{\mathcal{R}}(v) \leftarrow \Delta_{\mathcal{R}}(v) - 1$;

19: **return** $A$;

---

incrementally that can save overheads significantly. Specifically, for each RR set $R$ generated on $G(A)$, we just need to check whether $u$ should be inserted into $R$ after selecting a new edge $e = \langle u, v \rangle$, which suffices a correct update of $\mathbb{I}_R(\cdot)$. That is, we insert $u$ into $R$ if all of the following three conditions are met.

    (1) $R$ is not covered by $S$;
    (2) $v$ is in $R$;
    (3) $e = \langle u, v \rangle$ is live with a probability of $p_{u,v}$.

Otherwise, we simply retain the RR set $R$ with no changes. To explain, if all the aforementioned conditions are met, $u \in S$ with reach $v \in R$ after adding $e = \langle u, v \rangle$. Since the current $R$ is not covered by $S$ before adding $e$, we should insert $u$ into $R$ to ensure that $R$ is covered by $S$ after adding $e$. On the other hand, if $R$ is already covered by $S$, no further actions are needed as it always hold that $\mathbb{I}_R(\cdot) = 0$. Meanwhile, if $v \notin R$ and $R$ is not covered by $S$, $u$ still cannot reach any nodes in $R$ after adding $e$. Finally, if $e = \langle u, v \rangle$ is blocked with a probability of $1 - p_{u,v}$, the live-edge graph remains unchanged, so as to the RR set $R$.

## 4.3 Putting It Together

We instantiate Greedy with our newly developed estimator via RR sets in an accelerated generation way, namely **A**ugmenting the **I**nfluence of **S**eeds (AIS) with psudocode given in Algorithm 2.

    To obtain a multiplicative $\lambda$-error estimate $\widehat{\sigma}(A, S)$ of $\sigma(A, S)$, we employ the generalized stopping rule proposed by Zhu et al. [36]. That is, we keep generating a set $\mathcal{R}$ of random RR sets on $G$ until the coverage $\Lambda_{\mathcal{R}}(S)$ of $S$ in $\mathcal{R}$ exceeds $\frac{2(1+\lambda)(1+\frac{\lambda}{3})\log(2/\delta)}{\lambda^2}$ (Line 2), where $\delta \in (0, 1)$ is a predefined threshold on failure probability. Then, according to the result derived by Zhu et al. [36], $\frac{n}{|\mathcal{R}|}\Lambda_{\mathcal{R}}(S)$

is a $(\lambda, \delta)$-estimate of $\sigma(S)$, i.e.,

$$\Pr\left[(1 - \lambda)\sigma(S) \leq \frac{n}{|\mathcal{R}|}\Lambda_{\mathcal{R}}(S) \leq (1 + \lambda)\sigma(S)\right] \geq 1 - \delta. \quad (4)$$

Meanwhile, due to the law of large numbers, $\frac{n}{|\mathcal{R}|}\Lambda_{\mathcal{R}}(S \cup \{v\})$ is also a $(\lambda, \delta)$-estimate of $\sigma(S \cup \{v\})$ as $\Lambda_{\mathcal{R}}(S \cup \{v\}) \geq \Lambda_{\mathcal{R}}(S)$ for any $v$. Similarly, when we update RR sets incrementally, since $\Lambda_{\mathcal{R}}(S)$ increases, we can also produce a $(\lambda, \delta)$-estimate $\widehat{\sigma}(A, S)$ of $\sigma(A, S)$ for any $A$ via the updated $\mathcal{R}$. As a consequence, with a high probability, we have

$$\widehat{\sigma}(A \cup \{e\}, S) = p_{u,v}\widehat{\sigma}(A, S \cup \{v\}) + (1 - p_{u,v})\widehat{\sigma}(A, S)$$
$$\leq (1 + \lambda)\big(p_{u,v}\sigma(A, S \cup \{v\}) + (1 - p_{u,v})\sigma(A, S)\big)$$
$$= (1 + \lambda)\sigma(A \cup \{e\}, S).$$

Using an analogous analysis, with a high probability, we can get that $\widehat{\sigma}(A \cup \{e\}, S) \geq (1 - \lambda)\sigma(A \cup \{e\}, S)$. Now, we can devise an desired estimate $\widehat{\sigma}(A \cup \{e\}, S)$ of $\sigma(A \cup \{e\}, S)$.

    The Greedy algorithm selects edge $e$ with the largest $\widehat{\sigma}(A \cup \{e\}, S)$ in each iteration. By definition,

$$\widehat{\sigma}(A \cup \{e\}, S) = \frac{np_{u,v}}{|\mathcal{R}|}\Delta_{\mathcal{R}}(v) + \widehat{\sigma}(A, S).$$

Therefore, it is equivalent to choose $e$ that maximizes $p_{u,v}\Delta_{\mathcal{R}}(v)$ (Line 12). Before we select any edges, we compute $\Delta_{\mathcal{R}}(v)$ for each $v$ on the original RR sets (Lines 4–9). In each iteration, after selecting an edge $e^* = \langle u^*, v^* \rangle$, for each $R \in \mathcal{R}$, we perform a soft update if all the three conditions given in Section 4.2 are met (Lines 14–18). That is, instead of physically inserting $u^*$ into $R$, we set the value of the indicator function $\Pi(R)$ to 1 that indicates $R$ is now covered by $S$, and reduce $\Delta_{\mathcal{R}}(v)$ by 1 for every $v \in R$. Finally, after a subset $A$ of $k$ edges are selected, we terminate the algorithm and return $A$.

## 5 THEORETICAL ANALYSIS

In this section, we conduct a theoretical analysis of the proposed AIS algorithm, and make a comparison with MC-Greedy.

## 5.1 Analysis of AIS

We present the main theoretical result of AIS as follows.

THEOREM 5.1. *Algorithm 2 returns a $(1 - 1/e - \varepsilon)$-approximate solution to the IMA problem with a probability of at least $1 - \delta$, and runs in $O\big(\frac{k^2(m+n)\log(n/\delta)\sigma(\{v^\circ\})}{\varepsilon^2\sigma(S)} + k|E_C|\big)$ time, where $v^\circ$ is the most influential node in $G$.*

In what follows, we show the approximation guarantee and time complexity separately.

**Approximation Guarantee.** As shown in Section 4.3, AIS greedily selects the edge $e$ with the largest $\widehat{\sigma}(A \cup \{e\}, S)$ that is a multiplicative $\lambda$-error estimate of the augmented influence spread $\sigma(A \cup \{e\}, S)$ with a probability of at least $1 - \delta$. We examine at most $|E_C|$ edges in each iteration and there are $k$ iterations. By union bound, with a probability of at lest $1 - k|E_C|\delta$, all $\widehat{\sigma}(A \cup \{e\}, S)$'s are multiplicative $\lambda$-error estimates. Therefore, by scaling $\delta$ to $\frac{\delta}{k|E_C|}$ (Line 1 of Algorithm 2), according to Lemma 3.2, Algorithm 2 returns a $(1 - 1/e - \varepsilon)$-approximate solution to the IMA problem with a probability of at least $1 - \delta$.

**Table 2: Datasets ($K = 10^3$, $M = 10^6$, $B = 10^9$)**

| Name | $n$ | $m$ | Avg. Degree |
|---|---|---|---|
| GRQC | 5.2K | 14.5K | 5.58 |
| NetHEPT | 15.2K | 31.4K | 4.1 |
| Epinions | 75.9K | 508.8K | 13.4 |
| DBLP | 317K | 1.05M | 6.1 |
| Orkut | 3.1M | 234.2M | 76.3 |
| Twitter | 41.7M | 1.5B | 70.5 |

**Time Complexity.** For the generation of RR sets, according to (4), a total of $O\left(\frac{k^2 n \log(n/\delta)}{\varepsilon^2 \sigma(S)}\right)$ RR sets are generated, and generating each RR set takes $O\left(\frac{m}{n}\sigma(\{v^\circ\})\right)$ time [33]. Thus, the time complexity for generating RR sets is $O\left(\frac{k^2(m+n)\log(n/\delta)\sigma(\{v^\circ\})}{\varepsilon^2\sigma(S)}\right)$. For the selection of edge, a total of $O(k|E_C|)$ edges are examined. For the update of $\Delta_{\mathcal{R}}(v)$, the number of updates is upper bounded by the total size of the initially generated RR sets which is $O\left(\frac{k^2(m+n)\log(n/\delta)\sigma(\{v^\circ\})}{\varepsilon^2\sigma(S)}\right)$. Therefore, AIS runs in $O\left(\frac{k^2(m+n)\log(n/\delta)\sigma(\{v^\circ\})}{\varepsilon^2\sigma(S)} + k|E_C|\right)$ time.

With a reasonable assumption that the influence of any singleton node is smaller than that of the seed set $S$, i.e., $\sigma(\{v^\circ\}) \le \sigma(S)$, AIS runs in $O\left(\frac{k^2(m+n)\log(n/\delta)}{\varepsilon^2} + k|E_C|\right)$ time.

**Remark.** The state-of-the-art sampling method proposed by Guo et al. [14] takes $O\left((1+\frac{\mu}{n})\sigma(\{v^\circ\})\right)$ time for generating one random RR set, where $\mu = \sum_{\langle u,v\rangle\in E} p_{u,v}$ is that total propagation probability of $m$ edges. Applying it to the IMA problem, AIS runs in $O\left(\frac{k^2(\mu+n)\log(n/\delta)\sigma(\{v^\circ\})}{\varepsilon^2\sigma(S)} + k|E_C|\right)$ time. Assuming that $\sigma(\{v^\circ\}) \le \sigma(S)$, AIS runs in $O\left(\frac{k^2(\mu+n)\log(n/\delta)}{\varepsilon^2} + k|E_C|\right)$ time.

### 5.2 Comparison with MC-Greedy

Similar to the analysis for AIS, when generating $O\left(\frac{k^2 n\log(n/\delta)}{\varepsilon^2\sigma(S)}\right)$ Monte-Carlo simulations for estimating $\sigma(A, S)$, MC-Greedy achieves an approximation ratio of $(1-1/e-\varepsilon)$ with a high probability of $1-\delta$. For each Monte-Carlo simulation, it takes $O(m)$ time. Meanwhile, a total number $O(k|E_C|)$ of estimations are needed. Therefore, the time complexity of MC-Greedy is $O\left(\frac{k^3 mn|E_C|\log(n/\delta)}{\varepsilon^2\sigma(S)}\right)$. This implies that our AIS algorithm improves the time complexity of MC-Greedy by a multiplicative factor of $\frac{kn|E_C|}{\sigma(\{v^\circ\})}$. Notice that $|E_C|$ can be as large as $O(|S|n)$, which means we can improve the time complexity by a factor of $\frac{k|S|n^2}{\sigma(\{v^\circ\})}$.

## 6 EXPERIMENTS

### 6.1 Experiment Settings

This section evaluates the empirical performance of the proposed algorithm. All experiments are conducted on a linux machine with Intel Xeon(R) 8377C@3.0GHz and 512GB RAM. All algorithms are implemented by C++ with O3 optimization.

**Datasets.** We conduct our experiments on 6 datasets, the information of which is presented in Table 2. All datasets are publicly available [22, 24]. Among them, Twitter is one of the largest dataset

used in the field of influence maximization with up to millions of nodes and billions of edges.

**Parameters.** The number of seed users $|S|$ and the number of target edge set $k$ is both fixed as 50. The failure probability $\delta$ is 0.001. As this manuscript is targeted at the IC model, we use the IC model as the diffusion model. The propagation probability $p_{u,v}$ of each edge is set to be the inverse of $v$'s in-degree, which is a widely-adopted setting [5, 15, 18, 31–33]. To compare the quality of the solution sets, the expected influence spread is computed by the RR sets sampled in our algorithm, which guarantees a $(\varepsilon/k, \delta)$-estimation. In each of our experiments, we run each method for 5 times and report the average results.

In the first set of experiments, we randomly select 10000 edges to construct the candidate set $E_C$ and the seed set is selected at random. This setting brings convenience for the implementation of MC-Greedy, since when $|E_C|$ is large, MC-Greedy needs to traverse them all and simulate the diffusion process, which is prohibitive.

In the rest of experiments for larger datasets, to demonstrate the scalability of the proposed algorithm, the candidate edge set $E_C$ contains all possible edges, which means that the size of $E_C$ is $O(|S|n)$. And the seed set is selected by the IMM [32] algorithm, conforming the real scenario where we will select influential nodes as seed users.

**Candidate Edges Generation.** Every candidate edge $e = \langle u,v\rangle$ contains a seed node $u \in S$ and an ordinary node $v \in V \setminus S$. Considering both the probability of $u$ activating its out-neighbors and the probability of $v$ being activated, we heuristically assign the propagation probability $p_{u,v}$ as the mean value of $u$'s average out-degree and $v$'s average in-degree.

**Baselines.** We compare AIS with the following 7 baselines.

- RAND. Choose $k$ edges from $E_C$ uniformly at random.
- OUTDEG. Choose the edges that connect to the nodes with highest out-degree.
- PROB. Choose $k$ edges with highest probability.
- SINF. Using the RR sets generated by Algorithm 2, find top-$k$ nodes with highest marginal coverage, and select the edges pointing to them. The RR sets will not be updated after selection.
- AIS\P. AIS\P means AIS without considering the probability. Using the RR sets generated by Algorithm 2, for every round, select the edge pointing to the node with the highest marginal coverage. The RR sets are updated after each selection.
- AIS\U. AIS\U means AIS without updating the RR sets. Using the RR sets generated by Algorithm 2, for every round, compute the marginal gain by line 12 of Algorithm 2 but the RR sets will not be updated after each selection.
- MC-Greedy. The algorithm proposed in [9]. This algorithm is implemented with CELF [23] in our experiments.

### 6.2 Comparison with MC-Greedy

The first experiment set focuses on the comparison with baselines including MC-Greedy. Here we choose two relatively small datasets

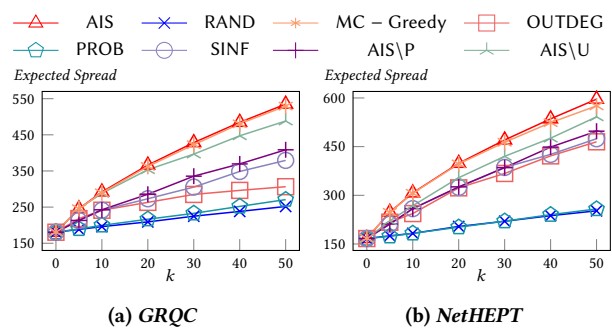

(a) *GRQC*      (b) *NetHEPT*

**Figure 1: Results on GRQC and NetHEPT ($\varepsilon = 0.5$).**

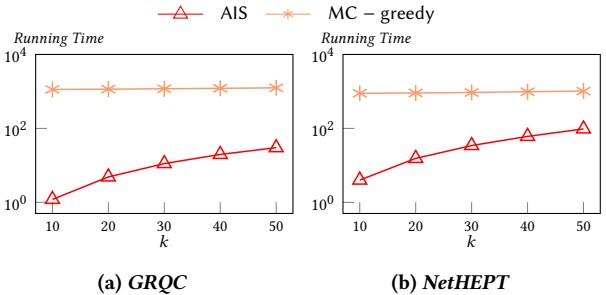

(a) *GRQC*      (b) *NetHEPT*

**Figure 2: Running time on GRQC and NetHEPT**

GRQC and NetHEPT to implement the algorithms. As the MC-Greedy algorithm suffers from high computation cost, if we traverse all $O(|S| n)$ candidate edges, the time cost is unacceptable. Therefore, here we fix the size of $E_C$ as 10000, and the number of Monte Carlo simulations is fixed to $r = 10000$. The seed users are selected uniformly at random. This is because if the influence of the seed users is already very large, it is harder to distinguish two edges as their marginal gain may be small. Then we need to increase the number of Monte Carlo simulations to obtain more accurate results, which is not acceptable. We adopt this setting just to show that our algorithms' efficacy is comparable to the state-of-the-art MC-Greedy algorithm [9].

From Figure 1, We can see that the performance of AIS and MC-Greedy are both effective, outperforming other heuristic baselines. This is consistent with our analysis, since they both follow the same framework and the only difference is the estimator. As for the running time, AIS is faster than MC-Greedy. MC-Greedy takes about one thousand seconds to return a solution while AIS only costs about half a minute. Figure 2 shows that although the setting is to the advantage of MC-Greedy, AIS is faster than MC-Greedy by about one order of magnitude when $k = 50$. Notice that the time cost of AIS increases with $k$, conforming our analysis in Section 5, while MC-Greedy's remains still. This is because we fix the value of $r$ for different $k$. If we want the MC-Greedy algorithm to obtain the same guarantee, the time cost will also increase with $k$ and be significantly larger than AIS's. In addition, when we select a more influential seed set, the time cost of MC-Greedy will also be higher. This is because in each iterations, we need to simulate the influence propagation process starting from these influential users. Due to the large influence of the seed set, it takes longer for the propagation process to stop. But it is not the case for our AIS

**Table 3: Running time (sec.) for different seeding strategies**

| Seeding Strategy | GRQC | | NetHEPT | |
|---|---|---|---|---|
| | RAND | IMM | RAND | IMM |
| MC-Greedy[1000] | 310.738 | 1131.529 | 304.410 | 1510.126 |
| MC-Greedy[2000] | 448.606 | 2177.472 | 326.002 | 3024.026 |
| MC-Greedy[3000] | 504.800 | 3517.178 | 404.653 | 4641.218 |
| MC-Greedy[4000] | 652.039 | 4861.590 | 434.475 | 6418.404 |
| MC-Greedy[5000] | 724.280 | 6196.576 | 640.814 | 7780.710 |
| AIS | 30.099 | 5.310 | 96.7855 | 14.766 |

algorithm. As analyzed before, AIS will cost less time when the expected spread of the seed set is larger. This claim is supported by the results in Table 3. We can see that when selecting influential seeds, our AIS algorithm is faster than MC-Greedy by two to three orders of magnitude.

## 6.3 Results on Larger Datasets

The second experiment set investigates the efficacy and efficiency of AIS on larger datasets. Here the seed users are selected by IMM algorithm [32], which fits the real scenario where we will select influential nodes as seeds and the candidate edge set contains all possible edges in $(S \times V)\backslash E$.

First, we run the baseline algorithms on these datasets and compare their performance with AIS's. Figure 3 shows the expected spread with different values of $k$. It is obvious that AIS consistently outperforms other heuristic baselines on all datasets. Notice that selecting the edges by out-degree or probability barely brings any benefit. This is because the solution set will contain a large number of edges pointing to the same node. Imagine that we successfully recommend a seed user $u$ to an ordinary user $v$ and $u$ can influence $v$ with large probability. Then the addition of the edge $\langle u, v \rangle$ will make other edges pointing to $v$ less beneficial. This issue is also observed in [8] and they empirically show that the influence spread will not be affected significantly even if we do not allow the addition of edges from different seeds to the same node. In contrast, SINF, AIS\P and AIS\U perform better. Notice that SINF and AIS\P basically achieve the same performance over four datasets. This is because their rationales are similar, which is to select influential nodes and avoid excess recommendations to the same target node. SINF simply selects top-$k$ influential nodes to connect to the seeds, and AIS\P selects the most influential node by the up-to-date RR sets in each iteration. Next we look into the performance of AIS\U. In NetHEPT, Epinions and Twitter, AIS\U presents considerable efficacy. However, it fails to give good solutions for DBLP and Orkut. After carefully investigating the solution sets, we find that AIS\U often provides excess recommendation to the same node since the marginal coverage array is not updated. That explains why sometimes the influence spread will not change much after adding new edges.

Figure 4 shows how the time cost changes with $k$. The running time of AIS increases with $k$, which is consistent with our analysis. Also, we observe that AIS, AIS\P and AIS\U share similar running time. The fact that AIS\U and AIS cost similar time indicates that the RR sets update process barely brings any extra cost. So although

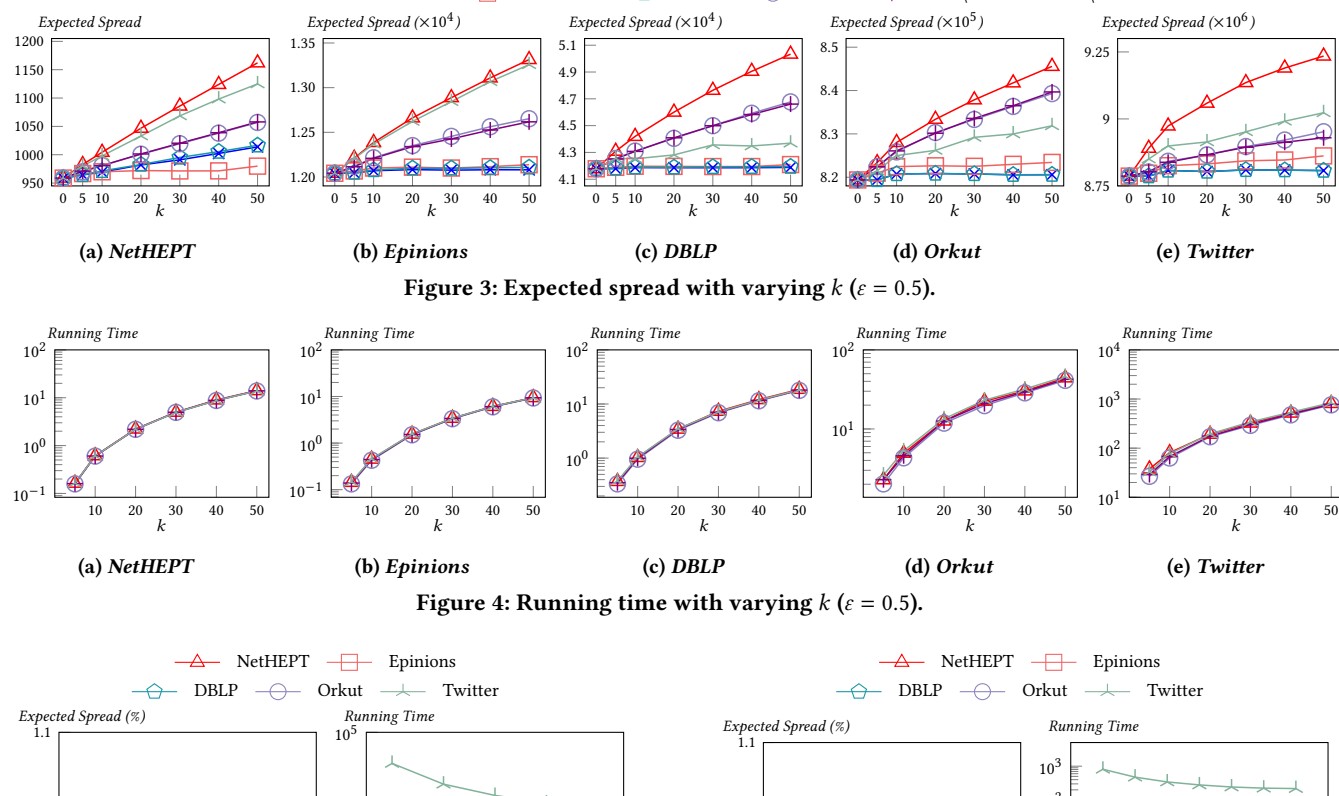

**Figure 3: Expected spread with varying $k$ ($\varepsilon = 0.5$).**

**Figure 4: Running time with varying $k$ ($\varepsilon = 0.5$).**

**Figure 5: Results with varying $\varepsilon$.**

**Figure 6: The effect of $\beta$ ($k = 50$).**

the we could only provide a loose bound on the time complexity for it, the RR sets update process will not cost much.

Then we investigate the relationship between running time and $\varepsilon$, the results of which are shown in Figure 5b. Intuitively, when the error term increases, the running time will decrease. For smaller error term, the speed of the increase of running time will be higher, which means we will pay larger cost to obtain more accurate results. To investigate whether such a cost is worthwhile, we conduct experiments to illustrate the relationship between the influence spread and $\varepsilon$. As shown in Figure 5a, the final influence spread barely changes with the increase of $\varepsilon$.

Despite the fact that we need to sample a mass of RR sets to maintain the theoretical guarantee, a natural question is whether we need such a large number of samples in practice. We set a parameter $\beta$ to control the number of RR sets to sample. That is, we set different values for $\beta$ and sample $\theta/\beta$ RR sets in our algorithm. Then we plot $\sigma(A, S)$ v.s. $\beta$. From Figure 6, we can see that reducing the number of samples does not have a significant impact on the solution quality, which makes AIS possible to apply to even larger datasets or deal with the scenarios where $k$ is large. However, notice

that the time cost decreases with different rates for different datasets and the adjustment of $\beta$ does not necessarily bring a linear time cost reduction. This is because when the sampling process is accelerated to certain extent, the bottleneck of the algorithm is on traversing the candidate edge set $E_C$.

## 7 CONCLUSION

In this paper, we study the IMA problem under the IC model that aims to recommend links connecting seed nodes and ordinary nodes to augment the influence spread of a specific seed set. We propose an efficient estimator for augmented influence estimation and an accelerated sampling approach, based on which, we devise AIS that can scale to large graphs with millions of nodes while maintaining the theoretical guarantee of $(1 - 1/e - \varepsilon)$. The experimental evaluations validate the superiority of AIS against several baselines, including the state-of-the-art MC-Greedy algorithm. As a future work, we plan to develop efficient algorithms for IMA under the liner threshold (LT) model. One can verify that Lemma 4.1 does not hold for the LT model. Hence, a key challenge lies in developing an efficient estimator under the LT model.

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

# A ADDITIONAL EVALUATION

From Figure 1b we can observe that the result of MC-Greedy is slightly inferior to that of AIS. This is because the number of samples for MC-Greedy is not enough. As we can see from Figure 7, when increasing the number of samples, the performance of MC-Greedy is also better. But when we increase $r$, the time expense will also increase a lot.

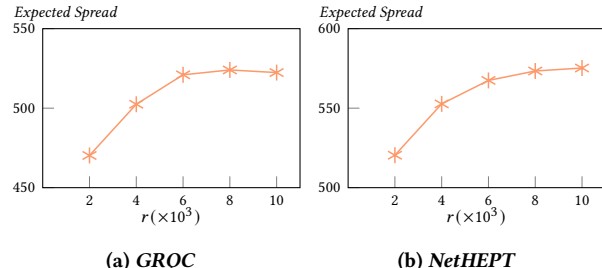

**(a) GRQC**     **(b) NetHEPT**

**Figure 7: Expected Spread v.s. $r$ on GRQC and NetHEPT**

Received DD MM YYYY; revised DD MM YYYY; accepted DD MM YYYY

