# OpenReview forum: "Link Recommendation to Augment Influence Diffusion with Provable Guarantees"
_ACM.org/TheWebConf/2024/Conference — TheWebConf24 Oral_

### Official Review · Reviewer_qiyo · 2023-11-21

**Novelty:** 4
**Technical Quality:** 5

**Review:**

Summary: This paper provides a linear time algorithm for the problem of {\it information maximization with augmentations}. Prior methods are not scalable as their runtime is super linear.  The authors have provided theoretical guarantees and they have conducted extensive experiments and demonstrated that their algorithms indeed beats the SOTA in both aspects.
This problem is introduced by D’Angelo et al [1] and it is a very natural extension of the influence maximization problem. Hardness and the submodularity property of the gain function are shown in [1] and here the authors propose a more efficient way of approximating the greedy choice using the {\it reverse sampling technique} of Borgs et al [2]. The reverse sampling technique has been introduced to obtain a linear time algorithm for the influence maximization problem and its usage for this problem is new.
While it is interesting to use reverse sampling technique here and the results solve an important problem and provide nice theoretical guarantees, I worry that this technique is not novel, since the problem is essentially influence maximization with a minor modification.

The paper is written clearly and all the claims are proven, it has a nice flow and provides a nice contribution.  My vote for this paper is a weak accept.

[1] Gianlorenzo D’Angelo, Lorenzo Severini, and Yllka Velaj. 2019. Recommending links through influence maximization. Theoretical Computer Science 764 (2019)
[2] Christian Borgs, Michael Brautbar, Jennifer Chayes, and Brendan Lucier. 2014. Maximizing social influence in nearly optimal time. SODA  2014.

Pros:
(1) interesting problem (2) claims are justified both theoretically and experimentally (3) clarity of the paper

Cons: (1) lack of technical novelty

Detailed comments for authors:

I think it is better to bring the summary of contribution after the related work section. Moreover,  present (in O tilde notation) the complexity prior works and SOTA together with your complexity. This way the reader can clearly see what is the theoretical improvement here.

line 286 where you explain the hardness of the problems: Isn't this proved in D'angelo et al's work. Also they show that 1-2/e approximation is NP-hard. Are we missing a 2 here?

You may cite the work related to link recommendation whose objective function is maximizing the information flow incorporating measures of fairness/diversity etc. For example:
Tsioutsiouliklis, Sotiris, et al. "Link Recommendations for PageRank Fairness." WWW 2022.
Florian Adriaens, Honglian Wang, Aristides Gionis Minimizing Hitting Time between Disparate Groups with Shortcut Edges KDD 2023
S Haddadan, C Menghini, M Riondato, E Upfal Repbublik: Reducing polarized bubble radius with link insertions WSDM 2021
Liwang Zhu, Qi Bao, and Zhongzhi Zhang. Minimizing polarization and disagreement in social networks via link recommendation. NeurIPS 2021.

------
Update after rebuttal phase:

I have read the responses to my review and other reviewers' input. I updated my vote accordingly. I think this is a good paper with solid results.

**Questions:**

I worry that the use of reverse sampling technique here is very natural as you essentially have the IM problem with new added edges. If I am missing something please correct me.

---

After reading rebuttal responses:

While this choice seems natural, I agree that using it in this context is novel and there are some major technical non trivialities that the authors resolve.

**Ethics Review Description:**

No issue

**Reviewer Confidence:**

4: The reviewer is certain that the evaluation is correct and very familiar with the relevant literature

**Scope:**

4: The work is relevant to the Web and to the track, and is of broad interest to the community

---

### Official Review · Reviewer_XZ1v · 2023-11-24

**Novelty:** 5
**Technical Quality:** 6

**Review:**

This paper proposes a new method AIS for the influence maximization with augmentation problem. Based on the classic greedy framework and reverse influence sampling (RIS) technique, the authors introduce two techniques to improve computational efficiency, i.e. an efficient estimator via RR sets and a fast generation process of RR Sets. Through theoretical analysis and extensive experimentation, the authors demonstrate that the proposed algorithm significantly improves computational speed while effectively ensuring the quality of computed results.

Pros:
1. The paper overall provides a high-quality exposition. It is well-organized and easy to follow.
2. This paper not only offers a solid theoretical analysis but also conducts comprehensive experiments to demonstrate the effectiveness of the proposed method. The method is sound and reasonable.
3. The authors conducted sufficient experiments on large-scale datasets to demonstrate that the proposed method not only exhibits outstanding performance (approaching the (1−1/e−𝜀)-approximation) but also shows a significant improvement in time efficiency.

Cons:
1. The statement of method in Section 4 is not easily understandable. More intuitive explanations could be included. For example, use some figures to better illustrate the main idea.
2. The proposed method relies on a fixed propagation model, which may result in its inability to adapt to complex real-world scenarios.
3. The baselines are not very convincing. In addition to some variants of the proposed solution, it only compares with the Greedy algorithm [9] of IMA task. The authors can consider the following baselines.
(1) Corò, Federico, Gianlorenzo D'Angelo, and Yllka Velaj. "Recommending links to maximize the influence in social networks.
(2) Coró, Federico, Gianlorenzo D’angelo, and Yllka Velaj. "Link recommendation for social influence maximization." ACM Transactions on Knowledge Discovery from Data (TKDD) 15.6 (2021): 1-23.

**Questions:**

This paper conducts extensive experiments on different-scale social graphs, which is very good. Are there real-world case studies to show that the proposed solution can improve the performance?

**Reviewer Confidence:**

4: The reviewer is certain that the evaluation is correct and very familiar with the relevant literature

**Scope:**

4: The work is relevant to the Web and to the track, and is of broad interest to the community

---

### Official Review · Reviewer_Kvsa · 2023-11-24

**Novelty:** 4
**Technical Quality:** 6

**Review:**

This work studies the problem of adding edges to a graph in order to maximize the expected influence spread from a given set of seed nodes. The main result is an efficient algorithm (linear in the size of the graph and of the set of candidate edges) that returns a solution provably within a factor of (essentially) $1-1/e$ of the optimum. Experiments confirm that in practice the proposed algorithm is efficient and (slightly) more effective than the competitors.

Overall this looks like a nice work. My only reservation is that the algorithm is almost just a combination of the greedy algorithm (select the edge yielding the largest increase in spread) and the reverse influence sampling algorithm (to estimate the spread increase of the edges). The only new ingredient is an observation that avoids re-running the reverse influence algorithm at each round of the greedy algorithm; in turns this brings the running time to be the sum of the greedy and RIS, rather than their product.

The manuscript is generally well written. However, I think the authors should make it clearer where the technical novelty lies here. Is it the observation above? If yes, and nobody noticed it before, then perhaps highlight it. Otherwise it looks "just" ordinary techniques combined together in an obvious way. In the same way highlight why previous work missed this combination of techniques (if that is true).

Minor issues:

- $E_C$ is repeatedly used (including in the contributions) without having been defined. Similarly, $k$ is used but is defined only in words. More in general, all the introduction repeatedly states bounds and claims without having clearly defined the problem and the notation (at least the fundamental one).

**Questions:**

See review.

**Ethics Review Description:**

--

**Reviewer Confidence:**

3: The reviewer is confident but not certain that the evaluation is correct

**Scope:**

4: The work is relevant to the Web and to the track, and is of broad interest to the community

---

### Official Review · Reviewer_oJ1D · 2023-11-28

**Novelty:** 4
**Technical Quality:** 5

**Review:**

This work proposes a scalable algorithm for the Influence Maximization with Augmentation (IMA, which aims to select new non-edges to be added to the graph) problem, by leveraging several connections between IMA and the classic IM problem (which aims to select seed nodes). By using an accelerated sampling approach, the authors propose an algorithm (AIS) which scales to large graphs while retaining the original submodular approximation guarantee for IMA.

The paper reads fluently, and is written well. The contributions are clear, namely a significantly more scalable algorithm for IMA than the vanilla greedy submodular optimizer.

minor comments:
- line 135:  "To the best of our knowl-
edge, this is the first algorithm that can be applied to large-scale
networks with theoretical guarantees." -> rewrite this sentence. First algorithm *for IMA*.
- line 140: is AIS an abbreviation? If so, include it the first time it appears in the paper.
- line 147: with a high probability of 1 − 𝛿 -> You can remove 'high'.

**Questions:**

Question: 1) IMA aims to add new non-existing edges between the seed and non-seed nodes. Similar edge augmentation problems between two not very well-connected groups have gained some recent research attention, see for example the papers 'Minimizing Hitting Time between Disparate Groups with Shortcut Edges' (KDD '23) and 'RePBubLik: Reducing Polarized Bubble Radius with Link Insertions' (WSDM '21), although these papers work with a different objective function (random-walk based). But loosely speaking, the goal is also to increase the network connectivity by adding new edges between two groups. Interestingly, IMA is submodular under the Ind. Cascade model, and so are the objective functions of the aforementioned papers (resp. super- and submodular). My guess would be that the algorithms proposed by those papers might also be useful for the IMA problem. Have you tested this, as another baseline in the experiments? Those papers also propose somewhat scalable sampling-based algorithms, although not as scalable as your methods.

2) What about the AIS running time when when every
singleton node’s influence is not smaller than the seed set?

**Ethics Review Description:**

no ethical issues

**Reviewer Confidence:**

3: The reviewer is confident but not certain that the evaluation is correct

**Scope:**

4: The work is relevant to the Web and to the track, and is of broad interest to the community

---

### Decision · Program_Chairs · 2024-01-22

**Decision:**

Accept (Oral)

**Comment:**

The reviewers all found this work to be a valuable contribution in proposing a sub-linear algorithm for the widely studied problem of influence maximization with augmentations. The reviewers' initial concerns have been sufficiently addressed, and we're happy to recommend acceptance of this work.